# Vision-Language Preference Optimization for Weakly Supervised Temporal Action Localization

## Abstract

Weakly supervised temporal action localization (WS-TAL) aims to localize actions in untrimmed videos using only video-level labels. Due to the absence of frame-level annotations, classification predictions during the initial training phase predominantly rely on the prior knowledge embedded in pre-trained video foundation models. However, the foundation model's inherent erroneous biases persist uncorrected during training, resulting in compounding error propagation throughout the learning process. To address this issue, we develop a dual-branch framework called Vision-Language Preference Optimization (VLPO) that enhances WS-TAL tasks through systematic integration with vision-language model. Our framework introduces two key components: (1) The Vision-Language Fine-Tuning (VLFT) branch, which effectively establishes a multimodal feature alignment mechanism through video-level supervision, conducts online adaptive fine-tuning on the vision-language features. This significantly enhances the semantic sensitivity of temporal localization under weakly-supervised conditions; (2) The Preference Driven Optimization (PDO) branch, through the predictive preferences provided by VLM, optimizes the traditional WSTAL framework and actionness learning at the snippet-level from both class-aware and class-agnostic perspectives, significantly enhancing the accuracy of action localization. Extensive experiments on WS-TAL benchmarks demonstrate that VLPO significantly outperforms state-of-the-art methods, showcasing its effectiveness in WS-TAL. The source code will be released upon acceptance.

## 1 Introduction

Temporal action localization (TAL) has garnered significant research attention due to its broad applications in video understanding. This task aims to simultaneously identify action categories and localize precise temporal boundaries within untrimmed videos. However, the labor-intensive nature of temporal boundary annotation creates substantial scalability barriers for real-world deployment.

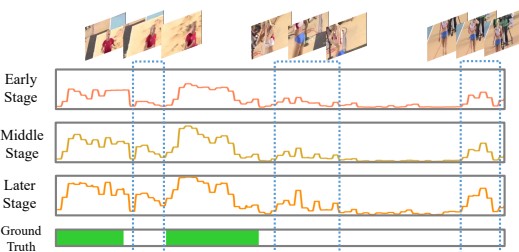

Figure 1: The CAS of baseline with actionness learning in different training stage.

To address this limitation, WS-TAL has emerged as a practical paradigm that only requires video-level category labels during training. Most existing WS-TAL methods [33; 13; 17; 9] first uniformly divide the input video into snippet units, each containing a fixed number of frames, and extract

snippet-level feature representations using pre-trained video foundation models (*e.g.,* I3D [2], Slow-Fast [5]). Subsequently, they adopt a Multi-Instance Learning (MIL) framework to learn and predict Class Activation Sequences (CAS), aggregate high-confidence segments to obtain video-level representations, and perform supervised learning based on video-level class labels.

However, due to the lack of precise snippet-level annotations, this approach suffers from significant learning bias. During the early stages of training, the WS-TAL model relies heavily on the prior knowledge of the pre-trained video foundation model for action localization, leading to a lot of misidentification that cannot be effectively corrected through video-level supervision.

Several methods [20; 15; 17] introduce an actionness learning branch to learn class-agnostic action information, thereby correcting the CAS predicted by the model backbone. This approach reduces action misjudgments caused by scene information and has achieved significant improvements. However, these methods have a fundamental limitation: the actionness learning branch shares the same source video snippet features with the backbone network, leading both to inherit the inherent biases of the pre-trained video foundation model. As a result, there are some action misjudgments in the backbone branch that cannot be corrected by actionness learning. As illustrated by the visualization results in Fig. 1, the baseline model with actionness learning exhibits persistent misclassification throughout the training process: the prediction confidence for some non-action snippets (*e.g.,* background snippets) not only fails to decrease with training but instead shows a monotonically increasing trend. This error accumulation effect ultimately results in a significant degradation of temporal localization accuracy.

In recent years, vision-language models (VLMs) pre-trained on large-scale cross-modal datasets have demonstrated strong transfer capabilities in tasks such as detection and segmentation. However, under the WS-TAL setting, it is not feasible to perform fine-grained snippet-level fine-tuning directly on VLMs. Moreover, directly applying VLM to the WS-TAL task in a zero-shot manner achieves an average mAP of only 14.9%, which is significantly lower than that of conventional WS-TAL methods (42.2%). In summary, this paper proposes a dual-branch weakly supervised learning framework based on VLM Preference Optimization, specifically addressing the fine-tuning of VLM under weak supervision and its optimization for the WS-TAL task. The core innovation lies in breaking traditional homogeneous feature constraints and achieving bias correction through cross-modal interaction.

Specifically, our VLPO framework includes two branches: (1) Vision-Language Fine-Tuning branch: We develop a Cross-Modal Anchored Feature Alignment (CM-AFA) module, which employs a anchoring strategy to stabilize cross-modal learning in weakly supervised settings. Then, We propose a Dynamic Selection Pooling (DSP) mechanism that utilizes preference matrix from CM-AFA and actionness to adaptively pool foreground and background snippets, leading to enhanced robustness in video representations. (2) Preference Driven Optimization branch: we propose the Preference Pseudo-label Generation (PPG) module to generate class-aware snippet-level supervision signals from VLM, thereby enhancing the localization accuracy of the backbone network. Furthermore, we design the Actionness Pseudo-label Refinement (APR) module, which combines temporal confidence calibration and contextual modeling to jointly optimize class-agnostic pseudo-labels.

Our contributions are summarized as three-folds: (1) We propose the VLFT branch to enable effective fine-tuning of VLM under weakly supervision. (2) We introduce the PDO branch, which optimize the WSTAL task by refining the prediction preferences of VLM from both class-aware and class-agnostic perspectives. (3) We conduct extensive experiments on WS-TAL benchmarks, demonstrating that VLPO outperforms existing state-of-the-art methods.

## 2 RELATED WORK

### 2.1 VISION-LANGUAGE PRE-TRAINING.

Recent years have witnessed significant advancements in vision-language models(VLMs) through cross-modal representation alignment and generative pre-training, substantially enhancing performance across multimodal tasks. Early approaches (*e.g.,* , VSE++ [4]) employed dual-stream architectures with cross-modal attention mechanisms to achieve preliminary image-text joint learning,

yet remained constrained by limited annotated datasets and single-task optimization. The integration of contrastive learning theory and large-scale pre-training paradigms has driven transformative progress. CLIP [23] established a transferable cross-modal semantic space through contrastive alignment of 400 million image-text pairs, demonstrating robust generalization capabilities in zero-shot classification and image-text retrieval tasks. BLIP [14] introduced a unified understanding-generation framework, jointly optimizing image-text matching, caption generation, and visual question answering via noisy data augmentation.

Temporal modeling extensions have further expanded VLMs' applicability. VideoCLIP [30] extended CLIP's contrastive framework to video-text alignment through multi-frame sampling and temporal attention modules, validating its effectiveness in video retrieval and action recognition. Subsequent work on VideoCLIP-XL [28] implemented targeted training for long-form textual descriptions, enhancing the model's comprehension capabilities.

## 2.2 WEAKLY SUPERVISED TEMPORAL ACTION LOCALIZATION.

WS-TAL is an approach that requires only video-level action class labels for supervision. Compared to the fully supervised approach, the labeling cost of the weakly supervised approach decreases substantially. Many works [33; 10; 15; 16] use a MIL framework to address this problem. Specifically, these works first perform classification at the snippet-level, then the top $k$ snippets with the highest scores in each category are aggregated to obtain the video-level predictions, and finally the model is optimized according to the video-level labels.

Actionness refers to the action attributes that deviate from the action category information. In ASL [20], the actionness branch is trained using the snippet-level class-agnostic pseudo-labels generated from the prediction results of the action classification branch. CoLA [33] computes actionness by summing class activation scores across categories.

Recent studies have extended WS-TAL from purely visual approaches to multimodal frameworks through the integration of language information. Li *et al.* [13] proposes a text-enhanced WTAL framework through text-segment mining and video-text completion. PVLR [19] proposes a human action-aligned probabilistic embedding framework integrating VLP knowledge with distribution contrastive learning enhancement.

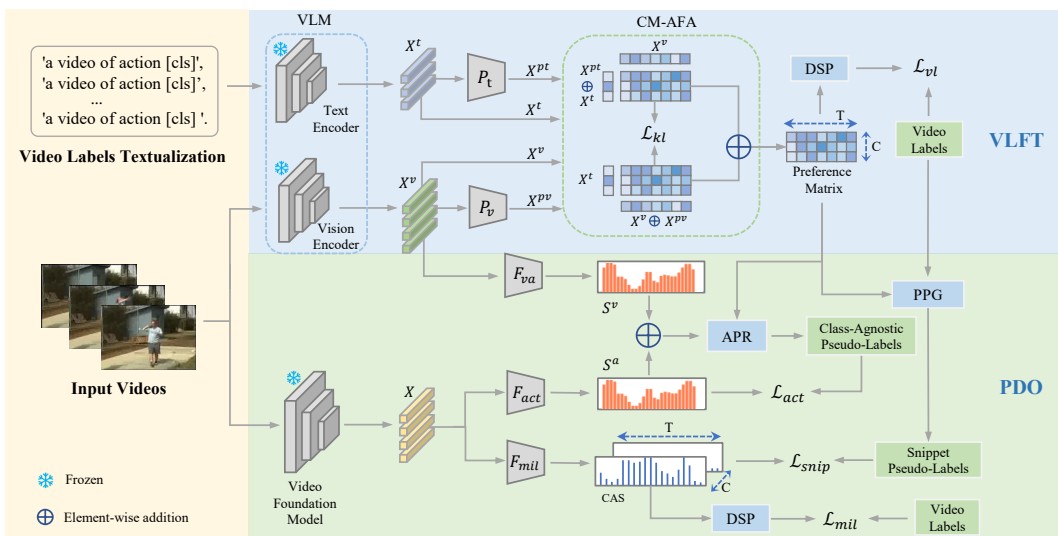

Figure 2: The framework of the proposed VLPO. It covers two parallel branches, named VLFT and PDO. In the VLFT branch, we freeze the pre-trained VLM while implementing multi-modal feature adaptation through two novel components: 1) CM-AFA module; 2) DSP module. These components collectively refine the VLM features for enhanced compatibility with WS-TAL task. In the PDO branch, we design dual optimization pathways using the vision-language preference matrix transferred from VLFT: 1) PPG module optimizes CAS learning; 2) APR module enhances actionness learning.

## 3 METHOD

This section introduce the different components of our proposed VLPO framework. As shown in Fig. 2, the VLPO framework consists of two branches, named Vision-Language Fine-Tuning (VLFT) branch and Preference Driven Optimization (PDO) branch respectively.

### 3.1 PROBLEM FORMULATION AND PRELIMINARIES

In this work, we define $\mathcal{V} = \{V_1, ..., V_N\}$ as a batch of data with $N$ videos and corresponding video-level action category labels are $Y_i = \{y_{i,1}, ..., y_{i,C}\} = \{0,1\}^C$ for $i$-th video, where $C$ means the number of category. In the inference stage, the model predicts all action instances in a video, then outputs a series of action instances with precise timestamps as $\{c, e, t_s, t_e\}$, where $c$ denotes the predicted action class, $e$ is the confidence score, $t_s$ and $t_e$ represent the start time and end time of the action instance.

Following most WS-TAL methods [22; 15; 16], we first divide an untrimmed video into a set of snippets, where each snippet contains 16 frames. Then we use the I3D network [2] pre-trained on the Kinetics-400 [2] to extract RGB and optical flow features for each snippet. The features of RGB and optical flow are 1024-dimension. For the $i$-th video with $T$ snippets, we concatenate RGB and optical flow features to obtain feature $X \in \mathcal{R}^{T \times 2048}$.

Give a feature $X$ extracted by the I3D, a temporal convolution layer followed by a ReLU function is applied to embed the features $X$ into the task-specific space. Specifically, we feed feature $X$ into the snippet-level classifier $F_{mil}$ to obtain the class activation sequence $\mathcal{A} \in \mathcal{R}^{T \times C}$, i.e., $\mathcal{A} = F_{mil}(X)$, where $F_{mil}$ represent the classifier. $C$ and $T$ is the number of action categories and the number of sampled snippets respectively.

To construct the classification scores of each action category at the video-level, we follow the mainstream approach of aggregating the highest $k$ scores across all segments for each action class and then taking their average. By applying the softmax function to the aggregated scores, we obtain the probability for each action category at the video-level: $p_c = \sigma(\frac{1}{k}\sum_{i=1}^{k} Topk(\mathcal{A}_{i,c}))$, where $p_c$ denotes the category probability at the video-level on the $c$-th category, $\sigma$ refers to the softmax function.

Following the previous works [22; 20], we apply the cross-entropy loss function between the predicted video-level action probability distribution $p_c$ and the ground truth $y_c$ to optimize the model. Specifically, we can formulate the classification cross-entropy loss as $\mathcal{L}_{mil} = -\sum_{c=1}^{C} y_c \log(p_c)$.

Due to the outstanding effectiveness of actionness learning [20; 15], we have introduced it in the baseline. we feed feature $X$ into the class-agnostic snippet-level classifier $F_{act}$ to obtain the actionness scores $S^a \in \mathcal{R}^{T \times 1}$.

Video snippets are divided into positive and negative sets, i.e., $\tau_p$ and $\tau_n$, where positive set $\tau_p$ contains $k$ snippets with the highest scores in $S^a$, and negative set $\tau_n$ has all other snippets. Following the ASL [20] model, we adopt actionness loss as:

$$\mathcal{L}_{act} = \frac{1}{|\tau_p|}\sum_{i \in \tau_p} \frac{1-(S_i^a)^q}{q} + \frac{1}{|\tau_n|}\sum_{i \in \tau_n} \frac{1-(1-S_i^a)^q}{q}, \tag{1}$$

where $0 < q \leq 1$ controls the noise tolerance and $i$ represents the $i$-th snippet.

Our baseline is optimized using the combined loss,

$$\mathcal{L}_{base} = \mathcal{L}_{mil} + \mathcal{L}_{act}. \tag{2}$$

### 3.2 VISION-LANGUAGE FINE-TUNING

Video VLMs such as VideoCLIP-XL [28] optimize model by aligning video-level vision semantics and text descriptions, without involving fine-grained information perception at the snippet-level. The inherent discrepancy between the pre-training objectives of VLMs and the requirements of temporal action localization tasks leads to significant performance degradation in direct application. In order to better adapt VLM to WS-TAL task, we design a CM-AFA module and a DSP module to fine tune the features extracted by VLM.

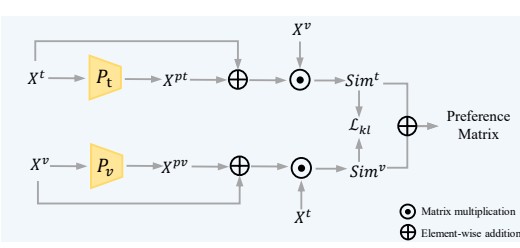 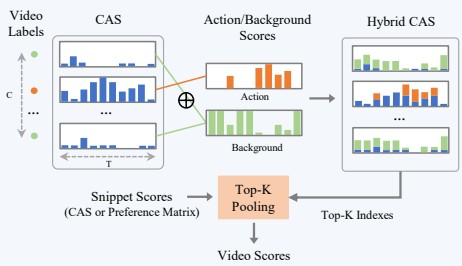

Figure 3: Details of the CM-AFA module.        Figure 4: Details of the DSP module.

**CM-AFA Module.** Concretely, for the visual stream, we utilize the VLM vision encoder to extract snippet-level features $X^v \in \mathcal{R}^{T \times 768}$. To bridge the domain gap between general visual representations and task-specific requirements, we implement a learnable linear projection layer $P_t$ that transforms feature $X^v$ into task-adaptive representation $X^{pv} \in \mathcal{R}^{T \times 768}$. While for the textual stream, we first convert the video label into text, such as "a video of action [cls]", where "[cls]" is the action category name, then feed them into the VideoCLIP-XL text encoder, to obtain textual features $X^t \in \mathcal{R}^{C \times 768}$. Similar to the operation of visual stream, we use a learnable linear projection layer to obtain projection text feature $X^{pt} \in \mathcal{R}^{C \times 768}$. Formally, $X^{pv} = P_v(X^v)$, $X^{pt} = P_t(X^t)$. To enhance training robustness through bidirectional cross-modal alignment, we establish anchor points using unimodal features and compute similarity scores (denoted as $M^t \in \mathcal{R}^{T \times C}$, $M^v \in \mathcal{R}^{T \times C}$) between each modality's native representation and its counterpart's projected features, as shown in Fig. 3. The specific process is as follows:

$$M^t = norm(X^v \cdot \left(\frac{(X^t + X^{pt})}{2}\right)^\top), \; M^v = norm(\frac{(X^v + X^{pv})}{2} \cdot (X^t)^\top). \qquad (3)$$

Compute the mean of $M^t$ and $M^v$ to obtain preference matrix $M = \frac{1}{2}(M^t + M^v)$. We formulate a symmetric KL divergence loss to enforce distributional consistency between cross-modal similarity measures. Formally,

$$\mathcal{L}_{kl} = \frac{1}{2}(D_{KL}(M^t \| M^v) + D_{KL}(M^v \| M^t)). \qquad (4)$$

**DSP Module.** Traditional MIL methods employ a coarse-grained Top-$k$ temporal pooling strategy to directly process CAS $\mathcal{A}$, which suffers from action-background semantic confusion and leads to insufficient discriminative feature learning. ASL [20], AICL [15] and our baseline incorporate CAS into actionness sequences without distinguishing between action categories. Consequently, only high-score snippets contribute effectively to foreground activation and background suppression, while low-score snippets receive insufficient optimization. To address this limitation, we design a Dynamic Selective Pooling module, as shown in Fig. 4. DSP enhances class activation via: (a) Target Action Enhancement: Selects class-specific sequences from the CAS that align with video labels and integrates them into the actionness sequences to enhance target action representation. (b) Non-target Action Suppression: Incorporates sequences from unrelated classes into the background score sequences to suppress non-target action snippets. The process is as follows:

$$\hat{\mathcal{A}} = \begin{cases} y_c \cdot \mathcal{A}_c + S, & \text{if } y_c = 1, \\ (1 - y_c) \cdot \mathcal{A}_c + (1 - S), & \text{otherwise,} \end{cases} \qquad (5)$$

where $S$ is the hybrid action score, obtained from Eq. 8. (c) Hybrid CAS Optimization: Top-$k$ indices (denoted as $idx$) are recorded for each class sequence in the hybrid CAS. Finally, video-level prediction scores are obtained by aggregating the optimized snippet-level scores based on $idx$, with supervision applied through video-level labels. Formally,

$$\mathcal{L}_{vl} = -\sum_{c=1}^{C} y_c \log(\hat{p_c}), \quad \hat{p_c} = \sigma(\frac{1}{k} \sum_{\forall i \in idx} M_{i,c}), \qquad (6)$$

where $M_{i,c}$ denotes the preference score of $i$-snippet on the $c$-th category, $\sigma$ refers to the softmax function.

## 3.3 PREFERENCE DRIVEN OPTIMIZATION

The PDO branch integrates vision-language preference with video-level supervision signals to optimize the WS-TAL task. This is achieved by designing two modules: PPG and APR, which respectively optimize the class-aware class activation sequences and the class-agnostic actionness score sequences through actionness learning.

**PPG Module.** The PPG module generates fine-grained snippet-level annotations to supervise the learning of class activation sequences. Specifically, we first normalize the preference matrix $M$ along the temporal dimension, denote as $\hat{M}$, then identify corresponding sequence scores through action categories annotated by video-level labels. We predefine a action threshold $\alpha_h$: for snippet with scores exceeding $\alpha_h$, their annotations inherit the video-level labels. If there are multiple action categories in the video-level labels, select the category with the highest corresponding sequence score as the annotation. Snippets below $\alpha_h$ are designated as background snippets and assigned uniform labels where each category receives an annotation value of $\frac{1}{C}$. Force the entropy of background snippets to be maximized, preventing them from achieving high scores on any action category. The sinppet pseudo-labels are denoted as $b \in \mathcal{R}^{T \times C}$. This mechanism establishes a collaborative supervision paradigm combining hard and soft labels, significantly improving discriminative representation learning in class activation sequences. Finally, we apply the cross-entropy loss function between the CAS $\mathcal{A}$ and sinppet pseudo-labels $b$,

$$\mathcal{L}_{snip} = -\frac{1}{T} \sum_{i=1}^{T} \sum_{c=1}^{C} b_{i,c} \log(\sigma(\mathcal{A}_{i,c})). \tag{7}$$

**APR Module.** The APR module employs a vision-language confidence calibration mechanism combined with temporal context modeling to jointly optimize class-agnostic snippet-level pseudo-labels through actionness learning. This approach effectively mitigates the label noise propagation issue inherent in conventional weakly supervised methods. Specifically, we first process the snippets features $X^v$ extracted by the vision encoder of the VLM through temporal convolution $F_{va}$ for temporal modeling, thereby obtaining the temporal action scores $S^v$. Then fuse the temporal action scores $S^a$ with $S^v$ to derive the hybrid action scores $S$, which is formulated as:

$$S = S^a + S^v, \quad S^v = F_{va}(X^v). \tag{8}$$

Following the baseline's actionness learning framework, we initialize positive/negative sample sets (denoted as $A_p, A_n$) via Top-$k$ method. Formally,

$$A_p = \{i | i \in Topk(S)\}, \quad A_n = \{i | i \notin Topk(S)\}. \tag{9}$$

Vision-language temporal confidence $S^M$ is computed as:

$$S^M = \hat{M}_c, \text{ if } y_c = 1, \tag{10}$$

where $\hat{M}$ is the normalized operation of $M$, as mentioned in PPG module. And $y$ is the video label. Construct vision-language preference sample sets,

$$B_p = \{i | S_i^M > \alpha_h\}, \quad B_n = \{i | S_i^M < \alpha_l\}, \tag{11}$$

based on predefined thresholds $\alpha_h$ and $\alpha_l$. Vision-language preference optimize positive/negative sample sets:

$$C_p = (A_p - B_n) \cup B_p, \quad C_n = \{i | i \notin C_p\} \tag{12}$$

We simplify Eq. 1 in the baseline as $\mathcal{L}_{act} = GCE(S^a, \tau_p, \tau_n, q)$, and the actionness loss after adding the APR module is:

$$\mathcal{L}_{act} = GCE(S^a, C_p, C_n, q) + GCE(S^v, C_p, C_n, q). \tag{13}$$

## 3.4 FINAL OBJECTIVE FUNCTION

The overall loss function we need to optimize is

$$\mathcal{L} = \mathcal{L}_{base} + \lambda_1 \mathcal{L}_{kl} + \lambda_2 \mathcal{L}_{vl} + \lambda_3 \mathcal{L}_{snip}, \tag{14}$$

where the $\lambda_1, \lambda_2, \lambda_3$ are the hyperparameters.

Table 1: Performance comparison with SOTA methods on the THUMOS14 dataset. This table reports the mAP values in IoU@{0.1:0.1:0.7}. The notation † denotes the incorporation of a vision-language model.

| Supervision | Method | Venue | mAP@IoU (%) | | | | | | | AVG | |
|---|---|---|---|---|---|---|---|---|---|---|---|
| | | | 0.1 | 0.2 | 0.3 | 0.4 | 0.5 | 0.6 | 0.7 | 0.1:0.7 | 0.3:0.7 |
| Full | G-TAD [31] | CVPR2020 | - | - | 66.4 | 60.4 | 51.6 | 37.6 | 22.9 | - | 47.8 |
| | RCL [29] | CVPR2022 | - | - | **70.1** | **62.3** | **52.9** | **42.7** | **30.7** | - | **51.7** |
| Weak | AICL [15] | AAAI2023 | 73.1 | 67.8 | 58.2 | 48.7 | 36.9 | 25.3 | 14.9 | 46.4 | 36.8 |
| | Li et al. [13] | CVPR2023 | - | - | 56.2 | 47.8 | 39.3 | 27.5 | 15.2 | - | 37.2 |
| | Ren et al. [24] | CVPR2023 | 71.8 | 67.5 | 58.9 | 49.0 | 40.0 | 27.1 | 15.1 | 47.0 | 35.6 |
| | Zhou et al. [36] | CVPR2023 | 74.0 | 69.4 | 60.7 | 51.8 | 42.7 | 26.2 | 13.1 | 48.3 | 38.9 |
| | PivoTAL [25] | CVPR2023 | 74.1 | 69.6 | 61.7 | 52.1 | 42.8 | 30.6 | 16.7 | 49.6 | 40.8 |
| | AFPS [17] | NN2024 | 73.5 | 68.8 | 60.8 | 51.3 | 41.0 | 27.5 | 16.5 | 48.5 | 39.4 |
| | ISSF [32] | AAAI2024 | 72.4 | 66.9 | 58.4 | 49.7 | 41.8 | 25.5 | 12.8 | 46.8 | 37.6 |
| | Hu et al. [9] | CVPR2024 | 74.1 | 69.2 | 60.0 | 49.8 | 41.1 | 28.0 | 15.1 | 48.2 | 38.8 |
| | SAL [16] | NN2024 | 76.3 | 71.6 | 63.7 | 54.2 | 41.8 | 29.0 | 17.9 | 50.6 | 41.3 |
| | Li et al. [13]† | CVPR2023 | - | - | 56.2 | 47.8 | 39.3 | 27.5 | 15.2 | - | 37.2 |
| | Ju et al. [12]† | CVPR2023 | 73.5 | 68.8 | 61.5 | 53.8 | 42.0 | 29.4 | 16.8 | 49.4 | 40.8 |
| | PVLR [19]† | MM2024 | 74.9 | 69.9 | 61.4 | 53.1 | 45.1 | 30.5 | 17.1 | 50.3 | 40.8 |
| | Zhang et al. [34]† | CVPR2025 | 74.3 | 69.8 | 61.8 | 52.3 | 43.0 | 30.8 | 16.6 | 49.8 | 40.9 |
| | VLPO (Ours)† | - | **78.9** | **75.2** | **68.7** | **60.0** | **45.5** | **32.3** | **19.8** | **54.3**$^{(+3.7)}$ | **45.3**$^{(+4.0)}$ |

Table 2: Performance on the ActivityNet 1.2 and 1.3 datasets. AVG is the averaged mAP at the thresholds {0.5:0.05:0.95}. The notation † denotes the incorporation of a vision-language model.

(a) ActivityNet 1.2

| Sup. | Method | Venue | mAP@IoU (%) | | | |
|---|---|---|---|---|---|---|
| | | | 0.5 | 0.75 | 0.95 | AVG |
| F. | SSN [35] | ICCV17 | 41.3 | 27.0 | 6.1 | 26.6 |
| W. | ASL [20] | CVPR21 | 40.2 | - | - | 25.8 |
| | CoLA [33] | CVPR21 | 42.7 | 25.7 | 5.8 | 26.1 |
| | DGCNN [26] | MM22 | 42.0 | 25.8 | 6.0 | 26.2 |
| | Li et al. [18] | MM22 | 41.6 | 24.8 | 5.4 | 25.2 |
| | DELU [3] | ECCV22 | 44.2 | 26.7 | 5.4 | 26.9 |
| | DDG-Net [27] | ICCV23 | 44.3 | 26.9 | 5.5 | 27.0 |
| | Ren et al. [24] | CVPR23 | 44.2 | 26.1 | 5.3 | 26.5 |
| | Hu et al. [9] | CVPR2024 | 45.1 | 27.7 | 5.5 | 27.6 |
| | SAL [16] | NN2024 | 48.5 | 31.4 | 7.1 | 30.8 |
| | Ju et al. [12]† | CVPR23 | 48.3 | 29.3 | 6.1 | 29.6 |
| | zhang et al. [34]† | CVPR25 | 48.3 | 30.1 | 6.8 | 30.1 |
| | VLPO (Ours)† | - | **56.0** | **34.5** | **7.6** | **34.8** |

(b) ActivityNet 1.3

| Sup. | Method | Venue | mAP@IoU (%) | | | |
|---|---|---|---|---|---|---|
| | | | 0.5 | 0.75 | 0.95 | AVG |
| F. | RCL [29] | CVPR22 | 51.7 | 35.3 | 8.0 | 34.4 |
| | DiffTAD [21] | CVPR23 | **56.1** | **36.9** | **9.0** | **36.1** |
| W. | ASM-Loc [7] | CVPR22 | 41.0 | 24.9 | 6.2 | 25.1 |
| | DGCNN [26] | MM22 | 37.2 | 23.8 | 5.8 | 23.9 |
| | Li et al. [13] | CVPR23 | 41.8 | 26.0 | 6.0 | 26.0 |
| | Ren et al. [24] | CVPR23 | 41.8 | 25.4 | 5.2 | 25.5 |
| | PivoTAL [25] | CVPR23 | 45.1 | 28.2 | 5.0 | 28.1 |
| | AFPS [17] | NN24 | 43.9 | 27.1 | 6.3 | 27.3 |
| | ISSF [32] | AAAI24 | 39.4 | 25.8 | 6.4 | 25.8 |
| | Li et al. [13]† | CVPR23 | 41.8 | 26.0 | 6.0 | 26.0 |
| | PVLR [19]† | MM24 | 43.6 | 27.4 | 6.5 | 27.4 |
| | VLPO (Ours)† | - | **50.7** | **32.8** | **7.8** | **32.9** |

# 4 EXPERIMENT

In this section, we conduct extensive experiments and visualize some results. For more experimental results, please refer to the **APPENDIX** A.

## 4.1 DATASETS AND EVALUATION

THUMOS14 [11] contains 200 videos in the validation set and 213 videos in the test set with 20 action categories. ActivityNet1.3 [1] contains 10,024 training videos, 4,926 untrimmed validation videos, and 5,044 videos for test whose action instance labels are withheld. This dataset contains 200 actions of different categories.

We use the mean Average Precision (mAP) with different temporal Intersection over Union (t-IoU) thresholds to evaluate the performance. Specifically, the t-IoU thresholds for THUMOS14 are [0.1:0.1:0.7], and [0.5:0.05:0.95] for ActivityNet1.2 and 1.3.

## 4.2 IMPLEMENTATION DETAILS

We apply I3D [2] as video foundation model and pre-trained VideoCLIP-XL [28] is selected as the VLM. Our model is trained using the learning rate 1e-4 for THUMOS14, and 1e-5 for Ac-

tivityNet1.3. Consistent with the baseline we set $T = 750, k = T/8$ for THUMOS14 and $T = 50, k = T/4$ for ActivityNet1.2 and 1.3. And we set $q = 0.6$ for all datasets. The new hyperparameters introduced by the VLPO framework remain consistent across all datasets, $\lambda_1 = 300, \lambda_2 = 0.1, \lambda_3 = 0.5, \alpha_h = 0.9, \alpha_l = 0.25$.

## 4.3 COMPARISON WITH SOTA METHODS

As demonstrated in Tab. 1, which presents TAL performance on the THUMOS14 dataset. Compared with weakly supervised methods, VLPO demonstrates substantial improvements over the current SOTA method SAL, with relative gains of 5.0%, 5.8%, and 3.7% at IoU thresholds 0.3, 0.4, and 0.5, respectively. Notably, our method achieves absolute improvements of 3.7% and 4.0% in average mAP over SAL, establishing new SOTA performance. Tab. 2 shows the results on ActivityNet1.2 and 1.3.

VLPO demonstrates notable superiority over other VLM-based methods [13; 19; 12; 34]. For distillation-based methodss [19; 12], due to the significant differences between VLM and WS-TAL tasks, they lack reasonable task-adaptive fine-tuning. In the VLPO method, the VLM features are fine-tuned through the VLFT branch and aligned with the WS-TAL task, enabling these features to be utilized more effectively and thus achieving a more pronounced performance improvement. Moreover, VLPO leverages a vision-language pre-training model without LLM components. This design yields substantial benefits in both model size and inference speed compared to approaches [34] based on MLLMs.

Table 3: Ablation studies on THUMOS14 datasets. "ACC" refers to the video-level classification accuracy. CM-AFA module takes effect through $\mathcal{L}_{kl}$ and $\mathcal{L}_{vl}$, while PPG module takes effect through $\mathcal{L}_{snip}$.

| Exp | $\mathcal{L}_{base}$ | $\mathcal{L}_{vl}$ | $\mathcal{L}_{kl}$ | $\mathcal{L}_{snip}$ | DSP | APR | ACC | AVG |
|-----|------|------|------|------|-----|-----|------|------|
| 1 | ✓ | | | | | | 90.0 | 42.4 |
| 2 | ✓ | ✓ | | ✓ | | | 93.3 | 50.6 |
| 3 | ✓ | ✓ | ✓ | ✓ | | | 92.4 | 51.0 |
| 4 | ✓ | ✓ | ✓ | | | ✓ | 90.5 | 45.0 |
| 5 | ✓ | ✓ | ✓ | | ✓ | ✓ | 99.0 | 50.6 |
| 6 | ✓ | ✓ | ✓ | ✓ | ✓ | | **99.5** | 52.1 |
| 7 | ✓ | ✓ | | ✓ | ✓ | ✓ | 99.1 | 52.9 |
| 8 | ✓ | ✓ | ✓ | ✓ | | ✓ | 90.5 | 47.8 |
| 9 | ✓ | ✓ | ✓ | ✓ | ✓ | ✓ | 99.0 | **54.3** |

Table 4: Comparison of different fine-tuning approaches for the VLFT branch on the THUMOS14 dataset.

| Fine-Tuning | mAP@IoU (%) | | | | AVG |
|-------------|------|------|------|------|------|
| | 0.1 | 0.3 | 0.5 | 0.7 | |
| Zero-Shot | 30.8 | 18.9 | 8.5 | 2.3 | 14.9 |
| Training-Free | 76.8 | 64.0 | 40.9 | 16.3 | 50.2 |
| CLIP-Adapter | 78.2 | 66.3 | 44.2 | 18.5 | 52.5 |
| CM-AFA | **78.9** | **68.7** | **45.5** | **19.8** | **54.3** |

## 4.4 ABLATION STUDY

**Effectiveness of each component.** To systematically validate the contribution of each component in our framework, we conduct comprehensive ablation studies on THUMOS14, with quantitative results summarized in Tab. 3. The baseline configuration (Exp 1) achieves 90.0% video-level classification accuracy (ACC) and 42.4% average (AVG) mAP. Exp 3 incorporates the CM-AFA and PPG modules and achieves 92.4% accuracy and 51.0% AVG mAP. This represents an 8.6% improvement in AVG mAP compared to the baseline. Exp 2, based on Exp 3 but excluding $\mathcal{L}_{kl}$, achieves 93.3% accuracy and 50.6% average mAP. Exp 4 integrates the CM-AFA and APR modules into the baseline, yielding an ACC of 90.5% and an AVG mAP of 45.0%. In Exp 5-8, we systematically remove individual modules from the complete VLPO framework to assess their respective contributions. In Exp 5, removing the PPG module causes a 3.7% reduction in AVG mAP. In Exp 6, excluding the APR module leads to a 2.2% decline in AVG mAP. In Exp 7, eliminating the $\mathcal{L}_{kl}$ results in a 1.4% deterioration in AVG mAP. In Exp 8, removing the DSP module severely degrades performance, with AVG mAP dropping by 6.5%. Exp 9 presents the performance of the complete VLPO framework, which outperforms the baseline by 11.9% in AVG mAP, underscoring the necessity of all integrated components.

Furthermore, our analysis reveals that the DSP module significantly enhances video-level classification accuracy. Without the DSP module, the maximum achievable accuracy is 93.3%. However, with the inclusion of the DSP module, the minimum observed accuracy is 99.0%. The DSP mod-

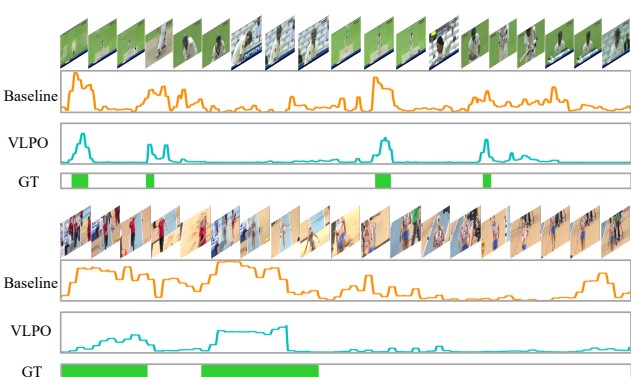

Figure 5: Qualitative results on THUMOS14.

ule effectively promotes the separation of action and background, enabling more precise capture of action snippets during temporal pooling.

**Effectiveness of different fine-tuning approaches.** To demonstrate the superiority of our CM-AFA module, we conduct ablation studies comparing three distinct approaches: (1) Zero-Shot: directly using VLM for snippet action prediction; (2) Training-free: using the VLPO framework without VLM features fine-tuning; (3) CLIP-Adapter: VLPO framework with VLM fine-tuning via method [6]. As shown in Tab. 4, the Zero-Shot approach achieves an AVG of merely 14.9%, significantly lower than other methods. This indicates that directly employing pretrained VLM for zero-shot prediction without integrating PDO branch or fine-tuning results in limited generalization capability for temporal action localization tasks. Particularly under strict evaluation thresholds (*e.g.,* IoU=0.7), the performance plummets to 2.3%, demonstrating the inadequacy of zero-shot approach in meeting high-precision localization requirements. By introducing the PDO branch without fine-tuning (Training-Free approach), the AVG substantially improves to 50.2%, marking a notable enhancement over the zero-shot baseline. This suggests that structural optimization alone, even without VLM fine-tuning, can significantly enhance the model's capacity to capture spatiotemporal features. The CLIP-Adapter method further elevates the AVG to 52.5%, yielding a 2.3% improvement over the Training-Free approach. CM-AFA emerges as the best method with 54.3% AVG, surpassing the Training-Free approach by 4.1%. It achieves the highest mean average precision (mAP) across all IoU thresholds (0.1-0.7), respectively. These results validate its robustness in weakly-supervised scenarios and highlight its pivotal role in comprehensive performance enhancement.

### 4.5 QUALITATIVE RESULTS

We visualize some examples of the detected action instances in Fig. 5. For each example, the top line represents the snippets of the video, the following three lines in order are the hybrid CAS $\hat{\mathcal{A}}$ generated by the baseline model and our VLPO, and the ground truth of the action in the video. The baseline framework incorporates actionness learning, but due to the inherent limitations of homologous features, numerous uncorrectable misjudgments persist. Our VLPO framework introduces VLM to optimize the prior biases existing in traditional frameworks, effectively rectifying the false positives observed in the baseline. The failure case of "LongJump" in Fig. 5 is mainly due to our VLPO focusing on the jump action while ignoring the action context, such as the run-up and rising. Weak supervision lacks snippet annotations, preventing models from learning run-up/rising are part of"LongJump". This stems from annotation limits (*e.g.,* vague action boundaries), not model flaws.

## 5 CONCLUSION

In this work, we propose a novel WS-TAL framework named VLPO, which introduces a VLM to assist in mitigating the prior bias issue inherent in conventional WS-TAL frameworks. The VLPO framework fine-tunes the VLM through its VLFT branch and transfers preference information to the WS-TAL task via the PDO branch. Our VLPO achieves significant improvements on three mainstream datasets, substantially outperforming existing SOTA methods.

## 6 REPRODUCIBILITY STATEMENT

We will open-source all experimental code within one week after the paper is accepted.

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

# A   APPENDIX

In this section, we first detail the specifics of the inference stage, followed by a comprehensive supplement of experimental results.

## A.1   INFERENCE STAGE

In the inference stage, we first calculate the video-level categorical probabilities $\hat{p}_c$ (Eq. 6) that indicates the possibility of each action class happened in a given video. Then we follow previous works [8] to set a class threshold $t_{class}$ to determine the action classes that would be localized in the video. Based on the identified action categories, we find the corresponding activation sequence from $\hat{\mathcal{A}}$ (Eq. 5). In practice, we use multiple action thresholds to process the activation sequence to enrich the proposal sets with different-level scales. After obtaining the action proposals, we utilize the $\hat{\mathcal{A}}$ to calculate the confidence score $e$ for each proposal using Outer-Inter Score. In the end, we remove the overlapping proposals using soft non-maximum suppression.

Table 5: Nosise robustness test of our VLPO. The AVG means average mAP under multiple t-IoU thresholds as {0.1:0.1:0.7} on THUMOS14.

| std | 0 | 0.1 | 0.2 | 0.3 |
|-----|------|------|------|------|
| AVG | 54.3 | 53.8 | 53.6 | 53.4 |

## A.2   NOISE ROBUSTNESS TEST

Our method operates under a weakly-supervised setting where snippet-level pseudo-labels used in training contain significantly noisy annotations. To test inference robustness, we add Gaussian noise with mean=0 to input features and evaluate different standard deviations (std), as shown in Tab. 5. Within a reasonable noise range, our model shows minimal performance fluctuation, demonstrating a certain degree of robustness.

Table 6: Sensitivity of $\lambda_1$. The AVG means average mAP under multiple t-IoU thresholds as $\{0.1:0.1:0.7\}$ on THUMOS14.

| $\lambda_1$ | 100 | 200 | 300 | 400 |
|---|---|---|---|---|
| AVG | 53.9 | 54.1 | 53.8 | 53.8 |

Table 7: Sensitivity of $\lambda_2$. The AVG means average mAP on THUMOS14.

| $\lambda_2$ | 0.05 | 0.1 | 0.2 | 0.3 | 0.4 |
|---|---|---|---|---|---|
| AVG | 53.7 | 54.3 | 54.1 | 53.8 | 53.8 |

### A.3 HYPERPARAMETER ROBUSTNESS EVALUATION

Robustness evaluations are performed on the five hyperparameters $(\lambda_1, \lambda_2, \lambda_3, \alpha_h, \alpha_l)$ introduced in the VLPO framework, and the results are presented in Tab. 6–10. The experimental results show that the values of the hyperparameters varied within reasonable limits, and the AVG mAP of the model fluctuated very little.

### A.4 ABLATION OF TEXT PROMPT

We conduct additional tests on various text prompts (including some with interference), as shown in the Tab. 11. As long as the prompt contains action categories ("[CLS]") from the dataset, there is no significant difference in performance.

### A.5 EFFICIENCY EVALUATION

In our method, we prefer using Vision-Language Pre-training (VLP) models such as VideoCLIP-XL [28], which does not include an LLM module and has a parameter count of only 400M, making it a lightweight model. We test VideoCLIP-XL and found that extracting features from a 169-second video takes only 0.36 seconds (on single RTX3090). This process can be fully parallelized with the video foundation model (I3D). Due to the integration of a multimodal large language model (MLLM) containing LLM components, the method introduced by [34] incurs a considerable overhead in terms of model size. Compared with traditional WSTAL methods, our approach introduces negligible additional inference time during the feature extraction stage. Excluding feature extraction models (I3D/VLM), we have compiled the parameter counts of some open-source method and their inference time on the full THUMOS14 test set, as concluded in the Tab. 12.

### A.6 QUALITATIVE ILLUSTRATION

We visualize some examples of the detected action instances in Fig. 6. (a) shows a video of the "CliffDiving" action, while (b) presents a video of the "BaseballPitch" action. Baseline model incorporates actionness learning, but due to its inherent prior bias, it predicts high action scores for many background snippets, leading to a large number of false positive proposals in the prediction results Our VLPO method significantly reduces false positive snippets in the prediction results by incorporating VLM to mitigate prior bias.

## B THE USE OF LARGE LANGUAGE MODELS

We use LLMs to polish the introduction section of this paper, making it more concise and understandable.

Table 8: Sensitivity of $\lambda_3$. The AVG means average mAP on THUMOS14.

| $\lambda_3$ | 0.1 | 0.3 | 0.5 | 0.7 | 0.9 |
|---|---|---|---|---|---|
| AVG | 54.2 | 54.3 | 54.3 | 53.9 | 53.6 |

Table 9: Sensitivity of $\alpha_h$. The AVG means average mAP on THUMOS14.

| $\alpha_h$ | 0.85 | 0.86 | 0.88 | 0.9 | 0.92 |
|---|---|---|---|---|---|
| AVG | 53.3 | 53.6 | 53.8 | 54.3 | 53.5 |

Table 10: Sensitivity of $\alpha_l$. The AVG means average mAP on THUMOS14.

| $\alpha_l$ | 0.15 | 0.2 | 0.25 | 0.3 | 0.35 |
|---|---|---|---|---|---|
| AVG | 54.0 | 54.0 | 54.3 | 54.2 | 53.8 |

Table 11: Ablation of text prompt. The AVG means average mAP on THUMOS14.

| Prompt | AVG |
|---|---|
| "[CLS]" | 53.9 |
| "find [CLS]" | 54.1 |
| "a video of [CLS]" | 54.1 |
| "a video of action [CLS]" | 54.3 |
| "an image of action [CLS]" | 53.9 |
| "avideoofaction[CLS]" | 54.1 |
| "a video of human action about [CLS]" | 54.0 |

Table 12: Efficiency evaluation.The time for inference on all samples on the test set is taken as the inference time.

| Method | Parameter Counts | Inference Time |
|---|---|---|
| CO2-Net [8] | 34.10M | 7.40s |
| AICL [15] | 6.30M | 7.19s |
| SAL [16] | 145.77M | 7.82s |
| VLPO (Ours) | 9.84M | 7.20s |

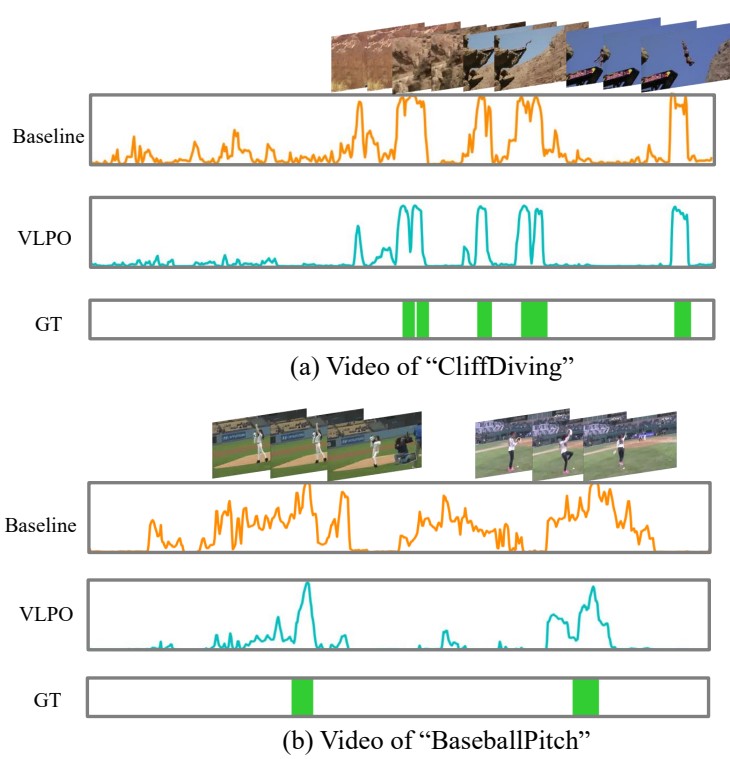

(a) Video of "CliffDiving"

(b) Video of "BaseballPitch"

Figure 6: Qualitative results on THUMOS14.

