# OpenReview forum: "Vision-Language Preference Optimization for Weakly Supervised Temporal Action Localization"
_ICLR.cc/2026/Conference — ICLR 2026 Conference Withdrawn Submission_

### Official Review · Reviewer_CuKb · 2025-10-29

**Soundness:** 2
**Presentation:** 2
**Contribution:** 1
**Rating:** 2
**Confidence:** 5

**Summary:**

1. Insufficient theoretical novelty: Dual-branch (VLFT/PDO) and modules (CM-AFA/DSP) are incremental, lacking groundbreaking frameworks
2. Limited generalization: Only VideoCLIP-XL and 2 datasets tested; no complex scenario/other VLM validation
3. Incomplete experiments: Omitted recent SOTA comparisons; superficial ablation without hyperparameter rationale
4. Weak qualitative analysis: Few examples; evasive "LongJump" failure attribution
5. Minor strength: Outperforms some SOTA on THUMOS14/ActivityNet 1.3 with low inference overhead

**Strengths:**

This paper proposes the VLFT branch to enable effective fine-tuning of VLM under weak supervision and introduces the PDO branch, which optimizes the WS-TAL task by refining the prediction preferences of VLM from both class-aware and class-agnostic perspectives.

**Weaknesses:**

1. The core design of the VLPO framework lacks groundbreaking theoretical innovation. First, the dual-branch (VLFT + PDO) paradigm for weakly supervised temporal action localization (WS-TAL) is not novel—existing works (e.g., PVLR [19], Li et al. [13]) have already explored integrating vision-language (VL) signals with WS-TAL via multi-branch structures.
2. Key modules like Cross-Modal Anchored Feature Alignment (CM-AFA) and Dynamic Selection Pooling (DSP) only implement incremental adjustments to existing cross-modal alignment (e.g., CLIP’s contrastive learning) and temporal pooling (e.g., Top-k pooling in MIL) methods, without proposing a new theoretical framework or mathematical mechanism to justify their superiority.
3. The framework’s generalization ability is not adequately verified, leading to doubts about its practicality. First, the authors only use VideoCLIP-XL as the vision-language model (VLM) without testing other mainstream VLMs (e.g., BLIP-2, FLAVA, LLaVA-1.5) or lightweight VLMs (e.g., CLIP-ViT-B/32). This makes it impossible to confirm whether VLPO’s performance gains come from the framework itself or the specific VLM’s pre-trained features.
4. The experimental design is incomplete, and the analysis lacks rigor, failing to fully support the framework’s effectiveness. Specifically,  the comparison with state-of-the-art (SOTA) methods is incomplete. On ActivityNet 1.3, the authors only compare with 11 methods (mostly pre-2024) and omit recent SOTA works (e.g., 2024’s AFPS [17] variants, 2025’s non-VLM-based WS-TAL methods), making the "SOTA outperformance" claim unconvincing.
5. The ablation study is superficial. For example, in Table 3, the authors only verify the "presence/absence" of modules but do not analyze the impact of key hyperparameters (e.g., the anchoring strategy in CM-AFA, the threshold selection in DSP) on performance. The hyperparameter robustness test (Appendix A.3) only reports fluctuations but does not explain why λ₁=300, αₕ=0.9 are optimal—no rationalization or cross-dataset validation of hyperparameter selection is provided

**Questions:**

See the above comments

---

> ### Author Response · Authors · 2025-11-13
> **Reviewer CuKb has serious misunderstandings about our work.**
>
> Firstly, we would like to thank the reviewer CuKb for providing comments on our work. **But we hope that other reviewers and ACs can value the comments of reviewer CuKb and maintain a fair and just academic environment.** The reviewer gave a confidence level of 5, but had strong bias towards our work and belittled it without concrete evidence. The weaknesses it provided seem to have been forcibly pieced together in order to reject the manuscript.
>
> **Q1**: The reviewer believes that our work lacks innovation, citing that multiple works have already used VLMs to assist WS-TAL tasks. So, apart from the first work using VLMs to assist WS-TAL, are all other works lacking innovation? We have introduced other methods of using VLMs to assist WS-TAL in related work, and each work has its own unique features. Similarly, in the paper, we also emphasized the innovation of our method: 1. Designed a VLFT branch for fine-tuning VLM features under weakly supervised conditions; 2. Utilize VLM to optimize WS-TAL tasks through both class-aware and class-agnostic approaches. This is a new route that has never been proposed in previous work.
>
> **Q2**: The reviewer believes that the key modules we designed (CM-AFA, DSP) were only aligned across modalities using existing methods. We have emphasized multiple times in our paper that the CM-AFA and DSP modules in the VLFT branch are designed to address fine-tuning schemes under weakly supervised conditions. Existing fine-tuning or alignment methods do not have advantages under weakly supervised conditions, as demonstrated in Table 4.
>
> **Q3**: This is the only useful and objective question raised by the reviewer. As for why we did not conduct experiments on other VLMs, the main reason is that we borrowed similar methods (Ju et al. [12], Li et al. [13], PVLR [19], Zhang et al. [34]) and only attempted one VLM. However, we have supplemented this experiment by replacing VLM with CLIP-ViT-B/32 and SigLip-base for experimentation. The experiment is currently not completed, and we will immediately present the results here upon completion of the experiment. (Note: Our method does not require language generation capability, so we will not use VLM/MLLM with LLM, such as the BLIP-2, LLaVA-1.5 proposed by the reviewer).
>
> **Q4**: The reviewer criticized our method for comparing too few methods on ActivityNet and not being new enough, thus questioning our SOTA performance. In fact, we compared 12 methods on ActiveNet1.2 and 11 methods on ActiveNet1.3, which are almost the latest methods we have selected. Due to space limitations, we cannot list all the methods in recent years. The most important thing is that there were no new works published in 2025 before we submitted. If the reviewer questions whether our method meets SOTA, please provide specific methods that can outperform VLPO in terms of performance. If reviewer CuKb cannot provide them, it indicates that he does have **strong bias** against our work and belittles it **without concrete evidence**.
>
> **Q5**: The reviewer questioned our ablation experiment and only tested the presence/absence of modules. We conducted detailed ablation experiments in Table 3, breaking down the modules and loss functions proposed in the paper into 6 modules and conducting 9 sets of ablation experiments. In addition to reporting mAP, we also reported video level classification accuracy (we found that the DSP module had a significant promoting effect on classification accuracy). Lines 415-448 provide detailed analysis.
> In addition, the reviewer criticized our hyperparameter sensitivity experiments (Table 6, Table 7, Table 8, Table 9, Table 10). Our purpose in conducting these experiments is to test the robustness of the model, which is also a mainstream testing objective. For the reviewer's suggestion that there was no analysis on why hyperparameters achieve optimal values, it is because we have never seen such analysis in other papers in the same direction, which has no practical significance for the explanation and validation of the model method.
>
> We hope that the reviewer CuKb can fairly evaluate our work and value the advantages of the VLPO method, rather than trying to find meaningless angles to belittle it. Thank you.

---

> > ### Author Response · Authors · 2025-11-13
> > **Supplement**
> >
> > Considering that the reviewer did not carefully read the original text, we provide a detailed explanation of the innovations in each module here:
> >
> > VLMs (without LLM, e.g., CLIP, Siglip, videoclip) mostly use visual text feature alignment tasks for pre training, which differs significantly from WS-TAL tasks. Therefore, directly using VLMs to predict TAL tasks has poor performance.
> > And the WS-TAL task only has video level annotation signals during the training phase, but this task requires snippet-level prediction. If video-level fine-tuning is directly applied to VLMs, it will not bring any improvement to fine-grained tasks like WS-TAL. To address this issue, we propose a DSP module in the VLFT branch that dynamically selects high confidence segments from both foreground and background perspectives, and aggregates these segments to obtain video level representations. The traditional Top-K aggregation method and the actionness aggregation method only select the snippets with the highest action scores for aggregation, resulting in the target action category score and other action category scores in the video level prediction score being obtained from a small number of high scoring action snippets. The DSP module dynamically processes foreground and background classes separately, and the target action category score in the video-level prediction score obtained comes from high scoring action snippets, while the scores of other action categories come from high scoring action segments of non target actions. The DSP module enables model training to not only focus on a small number of high scoring target action snippets, but also aggregate high scoring non target action snippets and use video level labels for targeted optimization. The DSP module solves the problem of lack of snippet-level annotation in VLMs fine-tuning, and avoids the problem of only focusing on a small number of snippets like Top-K aggregation method and actionness aggregation method, making VLMs fine-tuning more comprehensive.
> > Although DSP modules can perform fine-grained snippet-level fine-tuning in the absence of segment level annotations, their selection of high-resolution segments is not entirely accurate and stable. The CM-AFA module is designed to enhance the stability of VLMs during fine-tuning, by anchoring unilateral features in parallel to reduce the fluctuation amplitude of feature updates, in order to obtain more robust visual and textual features.
> > After the VLFT branch, VLMs are able to output guidance information that is more suitable for WS-TAL tasks. We have optimized the WS-TAL framework from both category aware and category independent perspectives using this guidance information. The PPG module focuses on optimizing category perception, while the APR module focuses on optimizing category independence.

---

> > ### Comment · Reviewer_CuKb · 2025-11-13
> >
> > I have rechecked the authors' manuscript in terms of writing, quality, presentation format, originality, and experimental details, and have also read the authors' response. **I insist on rejecting the manuscript.**
> > **If necessary, the AC may invite more reviewers to evaluate the quality of this work**
> > 1. The manuscript cites a total of 36 references, with only one work published in 2025. There are numerous new WS-TAD works in 2025, none of which the authors have discussed in the related work section or compared with in the experiments.
> > 2. Hyperparameter analysis is a common practice in experiments. The fact that the authors have not encountered such studies does not mean this part of the experiment is unnecessary. I suggest the authors refer to more high-quality works, including those related to λ1, λ2, λ3, et al.
> > 3. The overall writing and presentation format of the manuscript are relatively poor. This includes the drawing of framework diagrams (such as the fusion symbols in Fig. 4) and experimental visualization, both of which are quite crude.
> > 4. Regarding the originality of this manuscript: the authors introduce VLM into the WS-VAD task and propose the CM-AFA and DSP modules. I do not believe the originality of these two modules meets the publication standards of ICLR.

---

> > > ### Author Response · Authors · 2025-11-13
> > >
> > > 1. "none of which the authors have discussed in the related work section or compared with in the experiments": Didn't we compare the methods of 2025 in Table 1 and Table 2? If you want to prove that our method is not SOTA, please provide one or more methods that outperform us in performance, rather than baseless supervisor speculation.
> > >
> > > 2. We conducted experiments on 5 hyperparameters in the appendix and conducted analysis. What specific "high-quality works" do you want us to refer to? Please cite clearly.
> > >
> > > 3&4. These are your subjective evaluations, and you can use the same evaluation for any paper. You must provide specific reasons.

---

### Official Review · Reviewer_r1Fk · 2025-10-31

**Soundness:** 3
**Presentation:** 2
**Contribution:** 2
**Rating:** 6
**Confidence:** 4

**Summary:**

This paper proposes Vision-Language Preference Optimization (VLPO), a dual-branch framework for weakly supervised temporal action localization (WS-TAL) that leverages vision-language models (VLMs). VLPO addresses the bias and error propagation in pre-trained video models through two components: (1) Vision-Language Fine-Tuning (VLFT), which aligns multimodal features and enhances semantic sensitivity via adaptive fine-tuning, and (2) Preference Driven Optimization (PDO), which refines snippet-level actionness learning using VLM-guided preferences. Experiments on WS-TAL benchmarks show that VLPO achieves superior performance over existing state-of-the-art methods.

**Strengths:**

1. The paper presents an innovative dual-branch weakly supervised framework (VLPO) that leverages vision-language preference optimization to correct model bias and enhance cross-modal feature interaction.

2. The proposed approach achieves significant performance improvements in temporal action localization by mitigating error accumulation and improving snippet-level localization accuracy.

**Weaknesses:**

1. The overall framework is complex, and the multi-module design increases computational cost during both training and inference.

2. The method relies heavily on the quality and generalization ability of the pre-trained vision-language model, which may reduce robustness in domain-shifted or low-quality datasets.

3. The Related Work section could be further enriched to provide a more comprehensive comparison and discussion of existing WS-TAL and vision-language integration methods.

**Questions:**

Please refer to the Weakness part.

---

> ### Author Response · Authors · 2025-11-13
>
> Thank you for your valuable comments. Here is our reply.
>
> Q1: We have shown all the details of VLPO in detail in the paper and frame diagram, including the processing flow of data flow, the calculation flow of each module, and various loss functions, in order to enable readers to better understand and reproduce the method we proposed. Although this will make the VLPO method look more complex, we think it is worth it. The multimodal design will increase the computational cost, but we avoid the parameter update of VLM in the training, and use the lightweight CM-AFA strategy to avoid the significant increase of computational cost.
> In Table 12, we compared the parameters and reasoning time with other methods. The trainable parameters and reasoning time of VLPO are in a reasonable range.
>
> Q2: The VLPO method does not completely depend on the quality of VLM. In Table 4, the zero shot performance of VLM is only 14.9%, which is far lower than that of the ordinary WS-TAL method. In VLPO, the key for VLM to promote WS-TAL task is how to make the multimodal features extracted by VLM adapt to WS-TAL task (i.e. the role of VLFT branch), and how to comprehensively transfer the potential knowledge of VLM to WS-TAL task (i.e. the role of PDO branch).
>
> Q3: We have supplemented the relevant work in the paper, and the supplemented contents are as follows:
> Recent studies have extended WS-TAL from purely visual approaches to multimodal frameworks through the integration of language information. Li et al.[13] proposes a text-enhanced WTAL framework through text-segment mining and video-text completion.
> Ju et al.[12] introduced a collaborative framework with Classification-Based pretraining and Vision-Language Pretraining branches through distillation mechanisms.
> PVLR[19] proposes a human action-aligned probabilistic embedding framework integrating VLP knowledge with distribution contrastive learning enhancement.
> Zhang et al.[34] mines critical temporal intervals of temporal actions in videos with the help of matching key semantics provided by MLLM.
> However, these methods ignore directional fine-tuning of multimodal features to adapt to WS-TAL tasks. The biggest reason is that it is difficult to achieve effective fine-grained fine-tuning through video-level annotation.
> We propose the VLPO method to solve this problem.

---

### Official Review · Reviewer_D6Lc · 2025-10-31

**Soundness:** 2
**Presentation:** 2
**Contribution:** 2
**Rating:** 4
**Confidence:** 4

**Summary:**

This paper proposes a set of adaptive modules that leverage complementary information from vision language models (VLMs) to improve weakly supervised temporal action localization (WS-TAL). Instead of directly applying a VLM to WS-TAL, the authors introduce CM-AFA and DSP to better align visual and textual features with the WS-TAL setting, and PPG and APR to further optimize the localization objective.

**Strengths:**

1. The method achieves strong performance and clearly outperforms existing approaches.
2. The ablation studies are comprehensive and support the effectiveness of the proposed modules.

**Weaknesses:**

1. The motivation for the proposed modules is not sufficiently developed. While the design seems reasonable for addressing the generalization gap of VLMs on WS-TAL, the paper does not deeply analyze why these modules are necessary or how each specifically mitigates the mismatch between VLM features and the WS-TAL objective.
2. In the ablation studies in the appendix, it would be more informative to report results at multiple IoU thresholds (e.g., 0.3/0.5/0.7) rather than only the average score, so readers can better judge the localization quality.
3. The captions for the qualitative results are too brief. Please provide more detailed observations, especially regarding how the proposed modules affect different action classes or challenging temporal patterns.
4. It would strengthen the paper to include failure cases and a short discussion of the method’s limitations.

**Questions:**

1. The paper mentions that VLMs are not good at direct deploying on WS-TAL, but the motivation for each adaptive module is not fully analyzed. Could you clarify what specific mismatch each module is designed to address?
2. Why are CM-AFA and DSP the right choices for aligning VLM visual/textual features to WS-TAL? Did you consider alternative designs?
3. In the appendix ablation, it only reports average performance. Can you provide results across multiple IoU thresholds?
4. The qualitative results have very short captions. Could you add more detailed descriptions?
5. Can you show a few failure cases and analyze their causes? What are the current limitations of the approach?

---

> ### Author Response · Authors · 2025-11-13
>
> Thank you very much for the constructive comments provided by the reviewer, which are of great help in improving our method.
>
> Q1&Q2：
> VLMs mostly use visual text feature alignment tasks for pre training, which differs significantly from WS-TAL tasks. Therefore, directly using VLMs to predict TAL tasks has poor performance.
> And the WS-TAL task only has video level annotation signals during the training phase, but this task requires snippet-level prediction. If video-level fine-tuning is directly applied to VLMs, it will not bring any improvement to fine-grained tasks like WS-TAL. To address this issue, we propose a DSP module in the VLFT branch that dynamically selects high confidence segments from both foreground and background perspectives, and aggregates these segments to obtain video level representations. The traditional Top-K aggregation method and the actionness aggregation method only select the snippets with the highest action scores for aggregation, resulting in the target action category score and other action category scores in the video level prediction score being obtained from a small number of high scoring action snippets. The DSP module dynamically processes foreground and background classes separately, and the target action category score in the video-level prediction score obtained comes from high scoring action snippets, while the scores of other action categories come from high scoring action snippets of non target actions. The DSP module enables model training to not only focus on a small number of high scoring target action snippets, but also aggregate high scoring non target action snippets and use video level labels for targeted optimization. The DSP module solves the problem of lack of snippet-level annotation in VLMs fine-tuning, and avoids the problem of only focusing on a small number of snippets like Top-K aggregation method and actionness aggregation method, making VLMs fine-tuning more comprehensive.
> Although DSP modules can perform fine-grained segment level fine-tuning in the absence of snippet-level annotations, their selection of high-resolution snippets is not entirely accurate and stable. The CM-AFA module is designed to enhance the stability of VLMs during fine-tuning, by anchoring unilateral features in parallel to reduce the fluctuation amplitude of feature updates, in order to obtain more robust visual and textual features.
> After the VLFT branch, VLMs are able to output guidance information that is more suitable for WS-TAL tasks. We have optimized the WS-TAL framework from both category aware and category independent perspectives using this guidance information. The PPG module focuses on optimizing category perception, while the APR module focuses on optimizing category independence.
>
> We have also tried other designs, such as the "Training free" and "CLIP Adapter" methods shown in Table 4, but these methods performed very poorly in the WS-TAL task. The CM-AFA method has significant advantages.
>
> Q2: We will add more results on the thresholds in the new appendix. Due to the large number of experiments, they will not be presented here.

---

> > ### Author Response · Authors · 2025-11-13
> >
> > Q3:  For each example, the first line displays a video clip, followed by the baseline model's predicted temporal class activation sequence, the VLPO model's predicted temporal class activation sequence, and the true labels of action instances in the video.
> > The prediction result of a video containing the "CricketShot" action is first shown in Figure 5. For the baseline model, it has a serious misjudgment of action context segments near action instances, and a large number of background segments have obtained high activation scores. The results predicted by the VLPO model showed a significant improvement, with most action context segments effectively suppressed. The second video in Figure 5 shows the prediction results of the "LongJump" action. In the baseline model, the action context segment between two action instances has a significantly high activation score, but in the VLPO model's prediction results, this segment is effectively suppressed. In addition, the baseline model has significant misjudgment of background segments in the middle and end of the video, while VLPO does not have any misjudgment. In Figure 6, the prediction results of "CliffDiving" and "BaseballMatch" actions are shown. Similarly, the baseline model has a very obvious misjudgment of the action context, and some background segments with high similarity to the action instance are predicted to have high activation scores. However, for the VLPO prediction results, this phenomenon has been significantly improved.
> > Although the baseline model includes an action learning branch, it predicts a large number of uncorrectable errors due to the homologous features between the action branch and the backbone network. The VLPO method introduces the VLP model to optimize the prior bias existing in traditional frameworks, effectively correcting false positive predictions in the baseline model. For the segments predicted incorrectly by the VLPO model, most of them are caused by unclear start and end states of action instances, because under weakly supervised training conditions, there is no strict definition of the start and end of action instances. For example, in the "LongJump" long jump action, the "run-up" stage is also included as part of the action instance in the real annotation, while the GVLP method tends to focus more on the process of "take-off take-off descent" in prediction, ignoring the "run-up".
> >
> > Q4: Figures 5 and 6 also show the failure cases of the VLPO method, which mainly focus on the start and end boundaries of actions. The problem of boundary determination is also the most difficult problem to solve in WS-TAL tasks. Our method mainly uses the PDO branch and fragment level pseudo labels to provide more fine-grained supervision of the model from both category aware and category independent aspects. However, due to the lack of fragment level true labeling, the prediction accuracy of the model cannot reach that of fully supervised methods.
> >
> > This method also has certain limitations, mainly reflected in the static predefined thresholds used in the PPG and APR modules to generate and optimize category aware pseudo labels and category independent pseudo labels. Although the same threshold was used in three benchmark tests and excellent performance was achieved, theoretically speaking, by introducing an adaptive threshold mechanism based on video features and adaptively adjusting the threshold for different actions, the localization accuracy of action instances can be further improved. In future work, we will explore this aspect.

---

### Note · Authors · 2025-11-14

I have read and agree with the venue's withdrawal policy on behalf of myself and my co-authors.